# Deep Attentive Tracking via Reciprocative Learning

**Shi Pu**[1]    **Yibing Song**[2]    **Chao Ma**[3]    **Honggang Zhang**[1]*    **Ming-Hsuan Yang**[4]

[1]Beijing University of Posts and Telecommunications, Beijing, China
`{pushi_519200, zhhg}@bupt.edu.cn`
[2]Tencent AI Lab, Shenzhen, China
`dynamicstevenson@gmail.com`
[3]Shanghai Jiao Tong University, Shanghai, China
`chaoma@sjtu.edu.cn`
[4]University of California at Merced, Merced, U.S.A
`mhyang@ucmerced.edu`
https://ybsong00.github.io/nips18_tracking/index

## Abstract

Visual attention, derived from cognitive neuroscience, facilitates human perception on the most pertinent subset of the sensory data. Recently, significant efforts have been made to exploit attention schemes to advance computer vision systems. For visual tracking, it is often challenging to track target objects undergoing large appearance changes. Attention maps facilitate visual tracking by selectively paying attention to temporal robust features. Existing tracking-by-detection approaches mainly use additional attention modules to generate feature weights as the classifiers are not equipped with such mechanisms. In this paper, we propose a reciprocative learning algorithm to exploit visual attention for training deep classifiers. The proposed algorithm consists of feed-forward and backward operations to generate attention maps, which serve as regularization terms coupled with the original classification loss function for training. The deep classifier learns to attend to the regions of target objects robust to appearance changes. Extensive experiments on large-scale benchmark datasets show that the proposed attentive tracking method performs favorably against the state-of-the-art approaches.

## 1   Introduction

The recent years have witnessed growing interest in developing visual tracking methods for various vision applications. Visual attention plays an important role in facilitating tracking target objects in videos. For example, the state-of-the-art trackers based on discriminative correlation filters (DCFs) [21, 11] regress input features into a Gaussian response map for target localization. They often apply empirical spatial weights to input features to suppress the boundary effect caused by the Fourier transform. The spatial weights are generated by either a cosine [34] or a Gaussian [11] function. From the perspective of visual attention, we interpret these spatial weights as a specific type of attention maps. To improve the localization accuracy, the weights closer to the center regions of the input features are set to be larger. However, using these empirical attention maps limits the tracking performance when target objects undergo large movements. Meanwhile, deemphasizing non-central regions tends to downgrade the target response, leading to inaccurate localizations.

On the other hand, two-stage tracking-by-detection approaches first draw a set of proposals and then classify each proposal as either the target or the background. Visual attention has a great potential of facilitating learning discriminative classifiers. Existing deep attentive trackers [8, 27]

mainly use additional attention modules to generate feature weights. In other words, attention schemes are implemented by performing feature selection. Benefited from the end-to-end training, such attention schemes improve the tracking accuracy by strengthening the discriminative power of features. However, the feature weights learned in single frames are unlikely to enable classifiers to concentrate on robust features over a long temporal span. Moreover, slight inaccuracy of feature weights will exacerbate the misclassification problem. This requires an in-depth investigation on how to best exploit the visual attention of deep classifiers so that they can attend to target objects over time.

In this paper, we propose a reciprocative learning algorithm to exploit visual attention which advances the tracking-by-detection framework. Different from existing trackers using additional attention modules to weigh features, we directly train an attentive classifier. The training process consists of both a forward and a backward step. In the forward step, we feed an input sample into a deep tracking-by-detection network and compute the corresponding classification score. In the backward step, we take the partial derivative of this classification score with respect to the input sample along the direction from the last fully-connected layer towards the first convolutional layer. Note that, in the backward step, we do not update any network parameters. Instead, we take the partial derivative output of the first layer as the attention map. Each pixel value on this attention map indicates the importance of the corresponding pixel of the input sample to affect the classification accuracy. We exploit this map as a regularization term and add it in the loss function during training. The network parameters are updated following the conventional backward propagation scheme. As a result, the deep classifier learns to attend to the target regions and effectively eliminates background interference. In the test stage, the learned classifiers directly predict the classification score of each input sample for target localization. We validate the effectiveness of the proposed method on large-scale benchmark datasets. Our method shows favorable performance against the state-of-the-art approaches.

We summarize the main contributions of our work as follows:

- We propose a reciprocative learning algorithm to exploit visual attention within the tracking-by-detection framework.

- We use the attention maps as regularization terms coupled with the classification loss to train deep classifiers, which in themselves learn to attend to temporal robust features.

- We conduct extensive experiments on benchmark datasets where the proposed tracker performs favorably against state-of-the-art approaches.

## 2  Related Work

Visual tracking has been widely surveyed in the literature [42]. In this section, we mainly discuss the representative trackers and the related topic of visual attention.

**Visual tracking.**  The state-of-the-art visual tracking methods typically use a one-stage regression framework or a two-stage classification framework. The representative one-stage regression framework is based on discriminative correlation filters (DCFs), which regress all the circular-shifted versions of the input features into soft labels generated by a Gaussian function. By computing the spatial correlation in the Fourier domain as an element-wise product, the DCF trackers have received considerable attention recently due to the fast-tracking speed. Starting from the MOSSE tracker [4], extensions include kernelized correlation filters [20, 21], spatial regularization and multi-scale fusion [11, 13], CNN feature integrations [34, 38, 57, 26], and end-to-end predictions [48, 3, 46, 43]. Different from the one-stage regression framework, the two-stage tracking-by-detection framework mainly consists of two steps. The first step draws a sparse set of samples around the previously predicted location, and the second step classifies each sample as the target or the background. Considerable efforts have been made to improve the tracking-by-detection framework, including online boosting [17, 1], P-N learning [25], structured SVM [18, 36], CNN-SVM [22], random forests [56], multiple domain learning [35], adversarial learning [44], active tracking [32] and multiple object tracking [33]. In this work, we extend the two-stage tracking-by-detection framework by exploiting visual attention. Our deep classifier learns to attend to every discriminative region to separate target objects from the background.

**Visual attention.**  The visual attention starts from cognitive neuroscience, where the human perception focuses on the most pertinent subset of the sensory data. The visual attention scheme

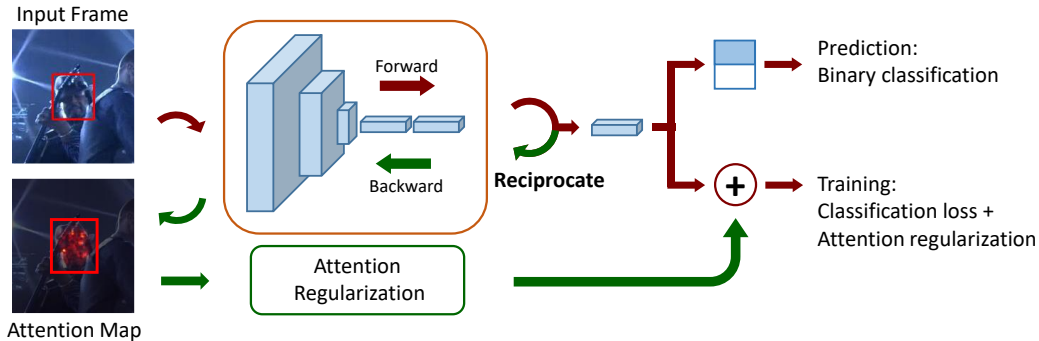

Figure 1: Overview of the proposed reciprocative learning algorithm. Given a training sample, we first compute its classification score in a forward operation. Then we obtain attention maps in a backward operation by taking the partial derivative of the classification score with respect to this sample. We use these maps as a regularization term coupled with the classification loss to train the classifier. In the test stage, no attention maps are generated. The classifier directly predicts the target location.

has been widely exploited for many computer vision applications, including image classification [40, 47, 23, 24], image caption [52], pose estimation [14], etc. In visual tracking, the spatial weights, which are widely used by DCF trackers to suppress the boundary effect, can be interpreted as one type of visual attention. Examples of such spatial weights include the cosine window map [4] and the Gaussian window map [11, 13]. Recently, a number of efforts [7, 8, 6, 27] have been made to exploit visual attention within deep models. These approaches emphasize attentive features and resort to additional attention modules to generate feature weights. The classifiers in these approaches are assumed not to be attentive. In this work, we propose a reciprocative learning process to learn an attentive classifier in the tracking-by-detection framework. We exploit attention maps as regularization terms coupled with the original classification loss for training classifiers. Our deep classifier with reciprocative learning in itself can attend to temporal robust features to improve the tracking accuracy.

# 3 Proposed Method

We propose a reciprocative learning scheme to activate the attentive ability of the classifier in the tracking-by-detection framework. Figure 1 shows an overview. Different from the existing attention models [47, 23, 7, 8, 6, 27] proposing additional modules to produce attention maps, we use the partial derivatives of the network outputs with respect to input images as attention maps. While some visualization methods [40, 39] use the attention maps to understand deep models, our method uses the attention maps to serve as regularization terms in the training stage to help the classifier attend to target regions robust to appearance changes. During testing, we directly use the classification scores to locate target objects. In the following, we first introduce how to incorporate attention maps in the original classification loss function. Then, we illustrate how the attention map gradually regularizes the classifier through reciprocative learning.

## 3.1 Attention Exploitation

We first present how we exploit the visual attention within the tracking-by-detection network. We denote by $I$ the input to the CNN tracking-by-detection network. The network outputs a vector of scores. Each element score indicates how likely $I$ belongs to a predefined class $c$. Given a specific input sample $I_0$, we use the first-order Taylor expansion [40] at a point $z_0$ to approximate the score function $f_c(I)$ as follows:

$$f_c(I) \approx A_c^\top I + B. \tag{1}$$

The point $z_0$ belongs to the deleted $\epsilon$-neighborhood of $I_0$ ($\epsilon \to 0$). The approximation (Eq. 1) holds true for any point in the $\epsilon$-neighborhood of $I_0$. Therefore, the derivatives of $f_c(I)$ at the points $z_0$ and $I_0$ are equal ($f_c'(z_0) = f_c'(I_0)$) as these two points are infinitely close. In Eq. 1, $A_c$ is the derivative

of $f_c(I)$ with respect to the input $I$ at the sample $I_0$:

$$A_c = \left. \frac{\partial f_c(I)}{\partial I} \right|_{I=I_0}. \tag{2}$$

Eq. 1 indicates that the output score of class $c$ is affected by the element values of $A_c$. In other words, the values of $A_c$ indicate the importance of the corresponding pixels of $I_0$ to generate the class score. As such, we can interpret $A_c$ as an attention map. For another specific input sample $I_1$, we again use Taylor expansion at a point $z_1$ to approximate $f_c(I)$. The point $z_1$ belongs to the deleted $\epsilon$-neighborhood of $I_1$. The new approximation holds true for any point in the $\epsilon$-neighborhood of $I_1$. Thus, the attention map $A_c$ corresponding to each input image sample is specific.

According to Eq. 2, we compute the partial derivative of the network output $f_c(I)$ with respect to the input $I$ at one specific sample $I_0$. This is achieved in two steps. First, we feed an input sample $I_0$ into the network and obtain the predicted score $f_c(I_0)$ in a forward propagation. Then, we take the partial derivative of $f_c(I)$ with respect to $I$ when $I = I_0$. According to the chain rule, this partial derivative is computed through backward propagation. We take the output of the first layer during backward propagation as the attention map $A_c$. We only select the gradients with positive values as they have clear contributions to the class scores with positive values. Thus, the attention map $A_c$ is always positive value and reflects how the network attends to the input sample $I_0$. Note that in the backward propagation, the network parameters are fixed without updating.

## 3.2 Attention Regularization

The tracking-by-detection framework usually defines the target object as the positive class and the background as the negative class to train a binary classifier. For each input sample $I_0$, we obtain two attention maps. One is the positive attention map (denoted by $A_p$) and the other is the negative attention map (denoted by $A_n$). For one positive training sample (labeled as $y = 1$), we expect the pixel values of $A_p$ related to target objects to be large. In comparison, the pixel values of $A_n$ related to target objects should be small. The attention regularization term for one positive sample can be formulated as:

$$R_{(y=1)} = \frac{\sigma_{A_p}}{\mu_{A_p}} + \frac{\mu_{A_n}}{\sigma_{A_n}}, \tag{3}$$

where $\mu$ and $\sigma$ are the mean and standard deviation operators for the attention maps. On the other hand, for one negative training sample (denoted as $y = 0$), we formulate its corresponding regularization term as:

$$R_{(y=0)} = \frac{\mu_{A_p}}{\sigma_{A_p}} + \frac{\sigma_{A_n}}{\mu_{A_n}}. \tag{4}$$

Using Eq. 3 and Eq. 4, we add the attention regularization terms into the original classification loss as:

$$\mathcal{L} = \mathcal{L}_{CE} + \lambda \cdot [y \cdot R_{(y=1)} + (1-y) \cdot R_{(y=0)}], \tag{5}$$

where $R_{(y=1)}$ and $R_{(y=0)}$ denote the regularization terms of the positive and negative training examples, respectively. The scaler parameter $\lambda$ balances the attention regularization terms and the cross-entropy loss $\mathcal{L}_{CE}$.

Eq. 5 shows how attention maps contribute to training the deep classifier. In addition to the classification loss, we incorporate the constraints from attention maps. For positive samples, we aim to increase the attention around the target object in two aspects. The first one is to increase the mean but decrease the standard deviation of $A_p$ so that the pixel intensity values are large and with small variance. The second one is to decrease the mean but increase the standard deviation of $A_n$ so that the pixel intensity values are small and with large variance. These two aspects reflect that the classifier learns to increase the true positive rates while decreasing the false negative rates. A similar intuition is shown in Eq. 4 where we decrease the false positive rates and increase the true negative rates of the classifier. As a result, the regularization terms help in increasing the classification accuracy by using the constraint from attention maps. This contributes to the classifier training process as the attention maps heavily influence the output class scores as shown in Eq. 1.

## 3.3 Reciprocative Learning

By incorporating the regularization terms in the loss function, the reciprocative learning algorithm is easy to implement. We resort to the standard backward propagation and chain rule. In each

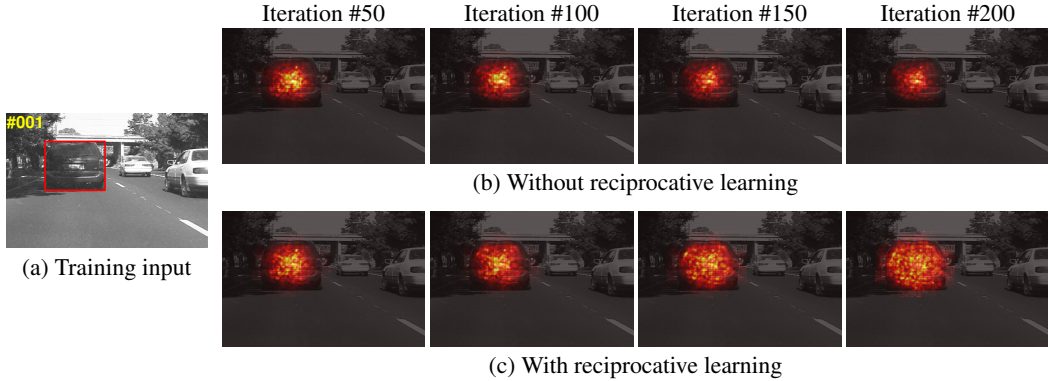

| Iteration #50 | Iteration #100 | Iteration #150 | Iteration #200 |

(a) Training input

(b) Without reciprocative learning

(c) With reciprocative learning

Figure 2: Visualization of attention maps on the *Car4* sequence [50]. With the proposed reciprocative learning algorithm, attention maps gradually cover the whole area of the target. The classifier thus attends to every region which can differentiate the targe from the background.

iteration of the classifier training, we compute the attention maps of every input training example. These attention maps reflect the attention of the classifier at the current status. Ideally, the classifier will selectively pay more attention to target objects than the background. As shown in Figure 2(b), the classifier without using attention regularization terms tends to focus on a limited number of discriminative regions. When target objects undergo large appearance changes, those limited regions are unlikely to represent target objects robustly in the whole video. With the use of the attention regularization, the classifier iteratively learns to attend to every region that can differentiate the target from the background. The classifier gradually focuses its attention on the whole target area. Figure 2(c) shows that the attention map only covers a subpart of the target region at the beginning of the training (i.e., Iteration #50). By means of reciprocative learning, the attention map gradually covers the whole target region. In the test stage, the attention regularization terms are not used. The classifier itself is able to attend to input samples.

## 4 Tracking Process

In this section, we discuss how to carry out the tracking task on a video. The proposed tracking algorithm does not require offline training. We mainly discuss three components regarding model initialization, online detection and model update.

**Model initialization.** In the first frame, we randomly draw $N_1$ samples around the initial target location. These samples are labeled as either positive or negative according to whether their intersection over union (IoU) scores with the ground truth annotations are greater than 0.5. We use $H_1$ iterations in the initialization step. For each sample in every iteration, we compute its loss using Eq. 5 and update the fully-connected layers accordingly.

**Online detection.** Given one frame in the video sequence, we first draw $N_2$ samples around the predicted location of the target in the previous frame. Then we feed each sample into our network. We select the candidate with the highest classification score and refine the target location using bounding box regression as in [16].

**Model update.** In each frame, we draw $N_2$ samples around the predicted target location. These samples are labeled as either positive or negative according to their IoU scores with the predicted bounding box. Then we use these samples to update the fully-connected layers using $H_2$ iterations in every $T$ frames.

We analyze how reciprocative learning contributes to the tracking-by-detection framework by visualizing the attention map, confidence map and the tracking results. The attention map shows how the network attends to the input image. The scores in confidence map indicate the probability of being the target object. We plot the predicted bounding boxes in red and the ground truth annotations in green. Figure 3 compares the visualization results with and without the reciprocative learning scheme. The state-of-the-art tracking-by-detection framework [35] is used as a baseline. We notice

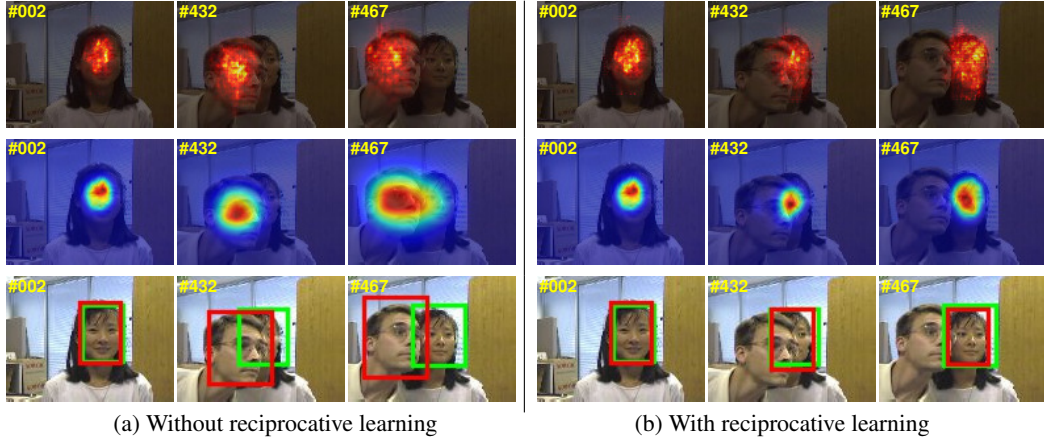

| (a) Without reciprocative learning | (b) With reciprocative learning |

Figure 3: Visualization of the prediction process with and without reciprocative learning on the *Girl* sequence [50]. From the first to the third row are: attention maps, confidence scores, and tracking results (in red). Ground truth annotations are in green. With the proposed reciprocative learning algorithm, the classifier pays more attention to temporal robust features and well differentiate the target from obstructions with similar appearance in Frame #432.

that at the beginning of the video sequence (Frame #2), the attention map, confidence map and the tracking results are almost the same between the baselines with and without reciprocative learning, respectively. This means that the attention map has not regularized the classifier well as it has not fully identified the target regions. However, along the tracking process, the reciprocative learning scheme helps the attention map cover the whole target region and thus strengthens the instance awareness. As a result, the network is able to produce a high confidence map on the target region. This helps the classifier differentiate the target from the background during occlusion even when the obstructions are highly similar (i.e., both are human faces) in Frame #432 and Frame #467. In comparison, the baseline without the reciprocative learning scheme drifts at the presence of occlusion as the classifier does not attend to temporal robust features.

## 5    Experiments

In this section, we first present the implementation details. Then we conduct ablation studies from two perspectives: 1) We investigate how the regularization terms contribute to learning discriminative classifiers. 2) To demonstrate the effectiveness of reciprocative learning, we compare with an alternative implementation using additional attention modules to generate feature weights. Finally, we evaluate our method on the standard benchmarks, i.e., OTB-2013 [50], OTB-2015 [51] and VOT-2016 [28]. We present more experimental results in the supplementary materials, and will make the source code available to the public.

**Implementation details.**    We use the same network architecture as in [35] to develop our baseline tracker. There are three fixed convolutional layers as the feature extractor and three learnable fully-connected layers as the classifier. The convolutional layers share the same weights with the VGG-M model [41]. The fully connected layers are randomly initialized and incrementally updated during the tracking process. Our goal is to update the parameters of the fully-connected layers to learn an attentive classifier. In the first frame, the number $N_1$ of samples is set to 5500. We train the randomly initialized classifier using $H_1 = 50$ iterations with a learning rate of 2e-4. In each iteration, we feed 1 mini-batch containing 32 positive and 32 negative samples into the network. In the online model update step, we fine-tune the classifier using $H_2 = 15$ iterations in every $T = 10$ frames with a learning rate of 3e-4. The network solver is stochastic gradient descent (SGD). During online detection, the number $N_2$ of proposals is set to 256. Our implementation is based on pytorch [37] and runs on a PC with an i7-3.4 GHz CPU and a GeForce GTX 1080 GPU. The average tracking speed is 1 FPS.

Table 1: Parameter sensitivity analysis of $\lambda$ on the OTB-2013 dataset. <span style="color:red">Red</span>: best. <span style="color:blue">Blue</span>: second best.

| $\lambda$ | 0 | 1 | 2 | 3 | 4 | 5 | 6 | 7 | 8 |
|---|---|---|---|---|---|---|---|---|---|
| DP | 0.911 | 0.917 | 0.936 | 0.941 | 0.940 | 0.944 | 0.938 | 0.929 | 0.911 |
| OS | 0.671 | 0.688 | 0.694 | 0.704 | 0.704 | 0.704 | 0.701 | 0.695 | 0.684 |

Table 2: Evaluation of baseline improvements using attentive features and attentive classifiers on the OTB-2013 dataset. <span style="color:red">Red</span>: best. <span style="color:blue">Blue</span>: second best.

| | Baseline | Baseline + Attentive Features | Baseline + Attentive Classifier (Ours) |
|---|---|---|---|
| DP | 0.911 | 0.932 | 0.944 |
| OS | 0.671 | 0.686 | 0.704 |

**Evaluation metrics.** We follow the standard benchmark protocols on the OTB-2013 and OTB-2015 datasets. We report the distance precision (DP) and overlap success (OS) rates under the one-pass evaluation (OPE). The distance precision rate measures the center pixel distance between the predicted locations and the ground truth annotations. We report the DP rate at a threshold of 20 pixels. The overlap success rate measures the overlap between the predicted bounding boxes and the ground truth annotations. We report the area-under-the-curve scores of the OS rate plots. In addition, we compute the average center distance errors in pixels, e.g., the center location error (CLE), and the overlap success rates at a threshold of 0.5 IoU ($OS_{0.5}$). On the VOT-2016 dataset, we use three evaluation metrics: expected average overlap (EAO), accuracy ranks (Ar) and robustness ranks (Rr).

## 5.1 Ablation Studies

We propose the reciprocative learning algorithm to extend the tracking-by-detection framework. In this section, we investigate how reciprocative learning contributes to learning discriminative classifiers. Note that we exploit attention maps as a regularization term in the loss function. The scalar parameter $\lambda$ in Eq. 5 makes a balance between the classification loss and the attention regularization term. We set $\lambda$ between 0 to 8 at an interval of 1 to evaluate the tracking performance on the OTB-2013 dataset. Table 1 shows that reciprocative learning consistently improves the baseline tracker (i.e., $\lambda = 0$) even the value of $\lambda$ is in a wide range. This affirms the effectiveness of the proposed reciprocative learning algorithm. Our method generally achieves top performing results when $\lambda$ is between 3 and 5. In the following experiments, we fix $\lambda = 5$ to report our tracking results.

Note that using attention maps as feature weights can also improve the tracking-by-detection framework as in [7, 8, 6, 27]. For fair comparison, we implement this strategy on top of our baseline by adding an attention module to generate feature weights. The classifier is learned on the attentive features with the original classification loss. Table 2 shows that using attentive features indeed improves the baseline tracking performance. However, the baseline tracker with reciprocative learning achieves much larger performance gains than that with the attentive feature scheme: 3.2% vs. 2.1% in DP, and 3.3% vs. 1.5% in OS. It is because our deep classifier itself learns to attend to the target object through attention regularization. While the attentive features learned in single frames are unlikely to be invariant to large appearance changes over a long temporal span.

## 5.2 Overall Performance

**OTB-2013 dataset.** We compare our method with 57 trackers. Among them 29 trackers are from the OTB benchmarks [50, 51]. The remaining 28 state-of-the-art trackers include ACFN [6], BACF [26], ADNet [53], CCOT [13], CREST [43], MCPF [57], CSR-DCF [31], SRDCFD [12], SINT [45], MDNet[35], HDT [38], Staple [2], GOTURN [19], KCF [21], TGPR [15], CNT [55], DSST [9], MEEM [54], RPT [30], SAMF [29], DLSSVM [36], BIT [5], SO-DLT [49], SCT [7], CNN-SVM [22], FCNT [48], HCF [34] and HART [27]. For presentation clarity, we only display the top 10 trackers. Figure 4 shows that our method generally performs well against state-of-the-art approaches on the OTB-2013 dataset with the distance precision and overlap success metrics. Specifically, our method improves the baseline MDNet by a large margin, which is not offline trained on auxiliary

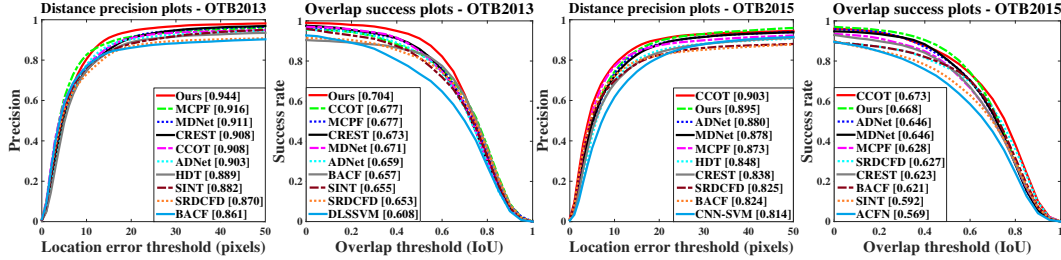

Figure 4: Distance precision and overlap success plots using the one-pass evaluation on the OTB-2013 and OTB-2015 datasets.

Table 3: Comparisons with the state-of-the-art trackers on the OTB-2013 and OTB-2015 datasets. Our tracker performs favorably against existing trackers in center location error (CLE), and the overlap success rate at a threshold of 0.5 IoU ($OS_{0.5}$). Red: best. Blue: second best.

|  | Trackers | Ours | CCOT | CREST | MDNet | MCPF | ADNet | BACF | SINT | SRDCFD | ACFN |
|---|---|---|---|---|---|---|---|---|---|---|---|
| CLE | OTB-2013 | 7.85 | 15.58 | 10.22 | 13.05 | 11.19 | 13.79 | 26.20 | 11.85 | 29.51 | 18.69 |
|  | OTB-2015 | 12.42 | 13.99 | 21.19 | 16.87 | 20.86 | 14.65 | 28.10 | 25.75 | 31.52 | 25.14 |
| $OS_{0.5}$ | OTB-2013 | 0.913 | 0.837 | 0.860 | 0.849 | 0.858 | 0.836 | 0.840 | 0.816 | 0.814 | 0.750 |
|  | OTB-2015 | 0.848 | 0.823 | 0.776 | 0.806 | 0.780 | 0.802 | 0.776 | 0.719 | 0.766 | 0.686 |

sequences for fair comparison. Table 3 shows the results in center location error and $OS_{0.5}$. The favorable results against state-of-the-art approaches demonstrate the effectiveness of our method in significantly reducing the average center distance error and increasing the tracking success rates.

**OTB-2015 dataset.** We compare our method with aforementioned trackers on the OTB-2015 dataset. Figure 4 shows that our tracker overall performs well against the state-of-the-art. While the top performing tracker CCOT achieves higher distance precision results, Table 3 shows that our tracker has a smaller center location error on all the benchmark sequences than CCOT. The overall favorable performance of our tracker can be explained by the fact that the proposed reciprocative learning algorithm strengthens the discriminative power of the classifier. The compared methods do not explicitly exploit visual attention. Our tracker does not perform as well as the top performing tracker CCOT in overlap success rate. It is because our tracker randomly draws a sparse set of samples for scale estimation. But CCOT crops the samples in a continuous space. We will explore this idea in the future work.

**VOT-2016 dataset.** We evaluate our method on the VOT-2016 dataset with the comparison to state-of-the-art trackers including Staple [2], MDNet [35], CCOT [13], EBT [58], DeepSRDCF [10], and SiamFC [3]. Table 4 shows that CCOT performs the best under the EAO metric. Our tracker overall performs comparably with CCOT, but much better than the others. The VOT-2016 report suggests that trackers whose EAO value exceeds 0.251 belong to the state-of-the-art. All these compared trackers including ours are thus state-of-the-art.

Table 4: Comparisons with the state-of-the-art trackers on the VOT-2016 dataset. The results are presented in terms of expected average overlap (EAO), accuracy rank (Ar) and robustness rank (Rr). Red: best. Blue: second best.

| Trackers | Ours | CCOT | Staple | MDNet | EBT | DSRDCF | SiamFC |
|---|---|---|---|---|---|---|---|
| EAO | 0.320 | 0.331 | 0.295 | 0.257 | 0.291 | 0.276 | 0.277 |
| Ar | 1.47 | 1.98 | 1.87 | 1.72 | 3.62 | 2.12 | 1.30 |
| Rr | 2.10 | 1.95 | 3.23 | 2.80 | 2.13 | 2.82 | 3.17 |

# 6   Concluding Remarks

In this paper, we propose a reciprocative learning scheme to exploit visual attention within the tracking-by-detection framework. For each input sample, we first compute the classification loss in a forward propagation, and take the partial derivatives with respect to this sample in a backward propagation as attention maps. Then we use attention maps as a regularization term coupled with the original classification loss function for training discriminative classifiers. Compared with existing attention models proposing additional modules to generate feature weights, the proposed reciprocative learning algorithm uses attention maps to regularize the classifier learning. Our classifier learns to attend to the robust features over a long temporal span. In the test stage, no attention maps are generated. Our classifier directly classifies each input sample. Extensive evaluations on the benchmarks demonstrate that our method performs favorably against state-of-the-art approaches.

## Acknowledgements

The work is supported in part by the Beijing Municipal Science and Technology Commission project under Grant No. Z181100001918005, Fundamental Research Funds for the Central Universities (2017RC08), NSF CAREER Grant No. 1149783, and gifts from NVIDIA. Shi Pu is supported by a scholarship from China Scholarship Council.

## Footnotes

*Honggang Zhang is the corresponding author.

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
