[Reviews · NeurIPS 2018]

Reviewer 1



This paper considers single-camera, single-target visual tracking where the target object can undergo large appearance changes. Attention maps helps visual tracking by selectively attending to temporally invariant motion patterns. The paper argues that existing tracking-by-detection approaches mainly use additional attention modules to generate feature weights. This paper proposes an algorithm consisting of feed-forward and backward operations to generate attention maps. The attention map is used as regularization terms coupled with the original classification loss function for training. The paper claims this is better than existing attention mechanisms used in visual tracking. The paper lacks novelty and significant contributions due to the following. 1. Visualization techniques have computed the sensitivity of input pixels to loss function. So the technique does not seem to be new. 2. Using the attention map as regularization terms is somewhat interesting. However, the results are not much better than existing attention mechanism. As stated in the paper, the baseline tracker with reciprocative learning gain over the baseline vs the baseline with attentive feature scheme gain over the baseline are: 3.2% vs. 2.1% in distance precision metric, and 3.3% vs. 1.5% in overlap success rates. 3. The extensive experiments comparing [33] with proposed attention map with respect to many other methods in the literature does not seem to be very relevant. There have been many papers on multi-target tracking, e.g. https://arxiv.org/pdf/1604.03635.pdf, Online Multi-Target Tracking Using Recurrent Neural Networks https://arxiv.org/pdf/1710.03958.pdf, Detect to Track and Track to Detect Multi-target tracking leverages state-of-the-art object detector. For two-stage object detectors, the proposal network acts like attention maps. For single-state object detectors, there is no need for attention. The authors might want to investigate whether attention map can reduce computation complexity in the context of multi-target tracking. In summary, there are two key issues. One is whether attention regulation is novel enough or not. The other is whether performance improvement is significant or not. Unfortunately, the rebuttal does not address either of them. The rebuttal claims performance gain of +1.1% in distance precision and +1.8% in overlap success over the state-of-the-art methods is significant. The rebuttal just reiterates that they are the first to integrates attention maps as a regularization term into the loss function for end-to-end learning CNNs.

Reviewer 2



A deep attentive tracking algorithm is proposed in this paper. Different from existing attentive tracking method which exploits attention map for feature weighting, the proposed method exploits the attention map as regularization to guide the network learning. The attention regularization is defined based on some measurements based on mean and standard deviation of the attention maps. Experiments are conducted on 3 benchmark dataset. Overall, this paper is well written. The description of the algorithm is clear and easy to understand. The literature review is comprehensive. The performance of the proposed tracking algorithm is excellent. However, there are still some weakness and issues which should be clarified: 1) The novelty of this paper is incremental. The proposed method is developed based on the MDNet framework. It seems that the only difference is that the proposed method further incorporate the attention regularization for backward propagation. 2) The regularization term seems a bit ad-hoc. Although the author has provided some intuitive explanation of the regularization, it seems lack of theoretical support. There are some other statistics which may be used to replace role of the mean and standard derivation in the regularization. Why they are not adapted in the regularization? For example, the median which is not sensitive to outlier value of data can be used to replace mean value. 3) The author claims that the proposed method can enable the classifier attend to temporal invariant motion patterns. It seems that no explanation is provided about what motion patterns mean in this paper. Although some figures show the evolvement of attention during training, no motion pattern is illustrated. In addition, some large variations may happen during the tracking process, such as out-plane-rotation, how can the proposed method ensure that the temporal motion invariant pattern can be found and the classifiers can attend to them? [POST-REBUTTAL COMMENTS] I have read the rebuttal and still have the concerns on the theoretical support for the regularization term. I keep my rating.

Reviewer 3



This paper proposes a new solution for visual tracking in the tracking-by-detection framework. The key idea is to use the derivative of the prediction scoring functions with respect to the image input as an attention map and regularize its mean and variance together with the classification loss. Traditional tracking-by-detection methods use separate attention module to generate feature weights, which may not well generalize. The proposed method could generalize well since the attention is naturally integrated into the detection model. Strong points: (1) The paper is well-written. (2) The idea of exploiting derivative w.r.t. the input image is interesting and seems significant. (3) The authors provide many experimental results and also video demo. However, I still have some major questions about it. The first one is that the reasoning in Section 3.1 is not rigorous. The first-order Taylor expansion of f_c(I) is A_c^T (I-I_0) + f_c(I_0), which means when you take derivative at I_0, and you want to evaluate f_c(I_0), then the first term disappears. Considering this, the reasoning after Eq.2 seems to be invalid. If we use another image I as input, and expand f_c(I) at I_0, then the computation of the derivative for all images should be w.r.t. I_0. The second question is, how would the regularizer (Eqs.3 and 4) behave if \mu_{A_p} and \mu_{A_n} are negative? For example, if \mu_{A_p} is negative, its absolute value would tend to zero if you are minimizing Eq.3. This is not what we want. Based on the rebuttal, I feel the paper is not clearly presented.